# A Novel Zero-Thermal-Quenching Red Phosphor with High Quantum Efficiency and Color Purity

Tianyang Zhao, Shiqi Zhang and Dachuan Zhu *

College of Material Science and Engineering, Sichuan University, Chengdu 610065, China;
2020223010033@stu.scu.edu.cn (T.Z.); 2021223015099@stu.scu.edu.cn (S.Z.)
* Correspondence: zhudachuan@scu.edu.cn

**Abstract:** In this paper, a series of $K_5La_{1-x}(MoO_4)_4$: $xSm^{3+}$ and $K_5La_{0.86}(MoO_4)_4$: $0.07Sm^{3+}$, $0.07Ln^{3+}$ (Ln = Sc, Y or Gd) red phosphors were prepared by calcining the mixed raw powders at 600 °C. Meanwhile, the composition and fluorescence properties of the phosphors, especially for the thermal stability, were analyzed in detail. The results indicate that the $K_5La_{1-x}(MoO_4)_4$: $xSm^{3+}$ phosphors can be effectively excited at 401 nm and emit red light with three main peaks at 561 nm, 600 nm and 646 nm, attributed to the $^4G_{5/2} \rightarrow {}^6H_{j/2}$ (j = 5, 7 and 9) energy transitions of the $Sm^{3+}$ ion respectively, among which the $K_5La_{0.93}(MoO_4)_4$: $0.07Sm^{3+}$ exhibits the highest intensity. The quenching mechanism is ascribed to the dipole-dipole interaction. $Ln^{3+}$ co-doping does not change the shape and peaking position of the excitation and emission spectra of $K_5La_{0.93}(MoO_4)_4$: $0.07Sm^{3+}$, but further increases the emission intensity in different degrees. Particularly, $K_5La_{0.86}(MoO_4)_4$: $0.07Sm^{3+}$, $0.07Gd^{3+}$ demonstrates a high quantum efficiency of 74.63%, a low color temperature (1753 K), and a high color purity of up to 99.97%. It is worth noting that all the phosphors have a good thermal stability, even a zero quenching phenomenon occurs, attributed to the electron traps confirmed by the TL spectrum.

**Keywords:** $K_5La_{1-x}(MoO_4)_4$: $xSm^{3+}$; $Ln^{3+}$ ions co-doping; zero-thermal-quenching; quantum efficiency





## 1. Introduction

Compared with traditional lighting methods, the new generation of LED lighting has a relatively longer life expectancy, produces less heat and consumes less energy with adjustable colors [1–4]. On the whole, it is in line with the current pursuit of low-carbon environmental protection and sustainable scientific development goals [5,6].

At present, using GaInN chips to excite YAG: $Ce^{3+}$ phosphors is still the most commonly seen solution of commercial lighting [7,8]. However, this scheme has some existing problems, such as high color temperature (CCT), low color rendering index (Ra) and blue-light hazards to the eyes [3]. Therefore, as one way to improve the color rendering ability of white LED, it is of practical significance to develop high-performance red phosphors or seek new excitation resource.

For now, the commercial red phosphor is mainly $Y_2O_3$: $Eu^{3+}$. However, due to its poor thermal stability [9] and invalidity to be excited by near-ultraviolet light [10], new red phosphors doped with $Mn^{4+}$, $Eu^{3+}$, $Sm^{3+}$ and $Pr^{3+}$ have been extensively explored, such as $BaLaMgTaO_6$: $Mn^{4+}$ [11], $BaMoO_4$: $Eu^{3+}$ [12], $LiCaGd(WO_4)_3$: $Eu^{3+}$ [13], $Ca_2LiScB_4O_{10}$: $Sm^{3+}$ [14], $Ca_3Y(AlO)_3(BO_3)_4$: $Sm^{3+}$ [15], $CsMgPO_4$: $Sm^{3+}$ [16], $Sr_3Ga_2Sn_{1.5}Si_{2.5}O_{14}$: $Sm^{3+}$ [17], $CaTiO_3$: $Pr^{3+}$ [18], $NaCaTiNbO_6$: $Pr^{3+}$ [19].

In a large number of phosphors with different matrix, rare earth doped molybdate has shown a great application potential in red phosphors for w-LED. For example, $La_2(MoO_4)_3$: $Eu^{3+}$ [20] phosphors have been found to have good red luminescence properties and an excellent thermal stability; $BaY_2(MoO_4)_4$: $Eu^{3+}$ [21] phosphors exhibit a stable red emission

with high quantum efficiency; $K_5La(MoO_4)_4$: $Eu^{3+}$ [22] phosphors show a high quantum efficiency and potential applications in fingerprint detection. In addition, studies have indicated that the phosphors could present excellent luminescent properties by using non-luminescent centers such as La, Y, and Gd, which have the same valence state and similar ionic radius as the matrix component or co-doped ions [23–25]. However, although previous studies have obtained some excellent phosphors, their comprehensive performance is still not good enough, and at the same time, the current theoretical study on the thermal stability of phosphors is not deep enough. It is still of practical significance to prepare excellent red phosphors with high luminous intensity, good thermal stability, high quantum efficiency and high color purity and to enrich their theoretical analysis.

In this work, a series of $K_5La_{1-x}(MoO_4)_4$: $xSm^{3+}$ and $K_5La_{0.86}(MoO_4)_4$: $0.07Sm^{3+}$, $0.07Ln^{3+}$ (Ln = Sc, Y or Gd) red phosphors were prepared by calcining the mixed raw powders at 600 °C. Meanwhile, the composition and preferable fluorescence properties of the phosphors, especially for the thermal stability, were analyzed in detail.

## 2. Experimental

### 2.1. Preparation of the Targets

Firstly, the raw materials including $K_2CO_3$(A.R.), $La_2O_3$(4N), $MoO_3$(A.R.), $Sm_2O_3$(4N), $Sc_2O_3$(4N), $Y_2O_3$(4N) and $Gd_2O_3$(4N) were dried and weighed proportionally in accordance with the designed chemical formula of the phosphors: $K_5La_{1-x}(MoO_4)_4$: $xSm^{3+}$ (x = 0.005, 0.01, 0.03, 0.05, 0.07, 0.10, 0.20 and 0.40) and $K_5La_{0.86}(MoO_4)_4$: $0.07Sm^{3+}$, $0.07Ln^{3+}$ (Ln = Sc, Y and Gd). After mixed thoroughly in an agate mortar for 20 min, the raw powders were transferred to a corundum crucible and calcined in air at 600 °C for 8 h. Finally, the products were cooled in the furnace and ground to obtain the target samples.

### 2.2. Characterization of Materials

The X-ray diffraction (XRD) patterns were obtained using Cu-K$\alpha$ radiation ($\lambda$ = 1.54056 Å) on the XRD-6000 (Hitachi Japan, Tokyo, Japan) in the range of 10–70° with a scanning step of 0.02°. The XRD results were refined by Rietveld method using GASA software (ver.gasa2full 5455). The microtopography, and the Mapping patterns of the samples were measured by the scanning electron microscope (SEM, JSM-7500, Tokyo, Japan) equipped with INCA X-Max50 (Oxford, UK). The excitation and emission spectra of the samples at different temperature were characterized using Hitachi F-4600 (Chiyoda, Tokyo, Japan) and its heating accessories (Orient KOJI TAP-02, Tianjing, China). The fluorescence lifetime and quantum efficiency of the samples were determined by FluoLog-3 (Horiba Scientific, Kyoto, Japan). The thermoluminescence (TL) spectra of the samples were recorded on TOSL-3DS (Guangzhou, China).

## 3. Result and Discussion

### 3.1. Phase Composition and Morphology

Figure 1a demonstrates the XRD patterns of $K_5La_{1-x}(MoO_4)_4$: $xSm^{3+}$ phosphors. As can be seen, all the locations and relative intensities of diffraction peaks fit well with the PDF card (#027-1363), and no other obvious secondary phases can be found, indicating that $K_5La(MoO_4)_4$-based phosphors are successfully prepared without preferential growth. The $K_5La(MoO_4)_4$ crystal belongs to the trigonal system with space group of R3m, where the La site is surrounded by eight oxygen atoms, forming the $[La/K1]O_8$ octahedral structure (Figure S1). It is worth noting that since the radius of $Sm^{3+}$ ion is smaller than that of $La^{3+}$ ion, the peaks slightly shift to a larger angle with the increase of doping concentration. Figure 1b shows the refined XRD patterns of $K_5La_{0.93}(MoO_4)_4$: $0.07Sm^{3+}$ phosphors, from which it can be found the calculation result is in good agreement with the experimental data, the $R_p$ and $R_{wp}$ are 9.13% and 7.39%, respectively. The information of atomic occupancy obtained from the refinement is listed in Table 1.

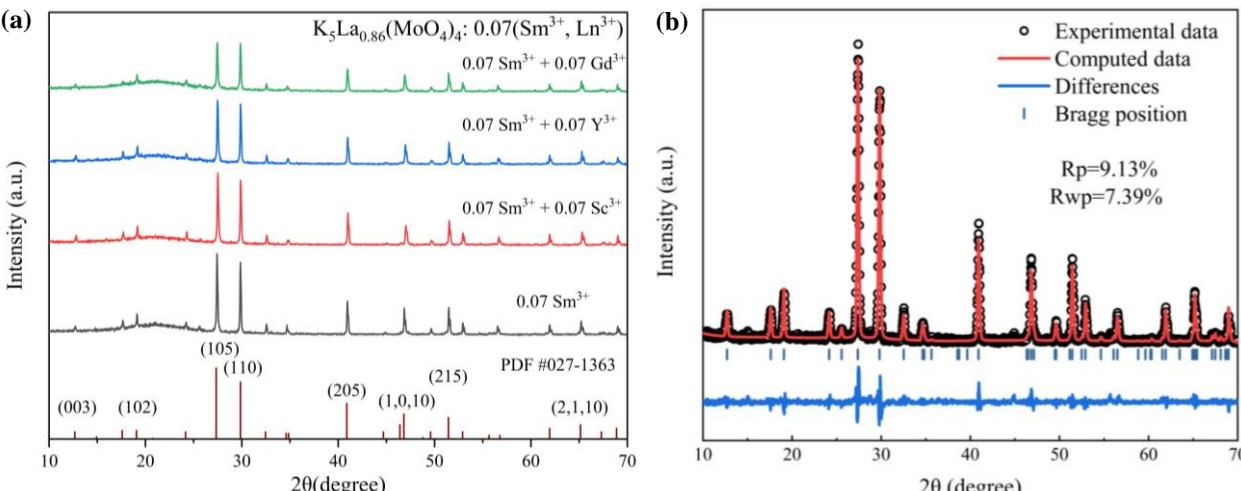

**Figure 1.** The XRD patterns of $K_5La_{1-x}(MoO_4)_4$: $xSm^{3+}$ phosphors (**a**), and the refinement pattern of $K_5La_{0.93}(MoO_4)_4$: $0.07Sm^{3+}$ phosphors (**b**).

**Table 1.** The atomic location parameters of refinement $K_5La_{0.93}(MoO_4)_4$: $0.07Sm^{3+}$.

| Label | Elem | Mult | x | y | z | Frac |
|-------|------|------|---|---|---|------|
| K1 | K + 1 | 3 | 0 | 0 | 0 | 0.418 |
| La1 | La + 3 | 3 | 0 | 0 | 0 | 0.495 |
| Mo1 | Mo + 6 | 6 | 0 | 0 | 0.400674 | 0.983 |
| K2 | K + 1 | 6 | 0 | 0 | 0.195013 | 1 |
| O1 | O − 2 | 18 | −0.043738 | 0.043738 | 0.318691 | 0.262 |
| O2 | O − 2 | 36 | −0.1855 | 0.1509 | 0.4173 | 0.13 |
| O3 | O − 2 | 18 | −0.144455 | 0.144455 | 0.429581 | 0.569 |

Figure 2 presents the SEM microtopography (a) and particle size distribution (b) of $K_5La_{0.93}(MoO_4)_4$: $0.07Sm^{3+}$ phosphors. According to the results of particle size analysis, it is observed that the particle sizes of phosphors are mainly distributed in 5–20 μm, and the average particle size is calculated to be 10.49 μm. It could be seen from the Mapping patterns in Figure S2 that $Sm^{3+}$ ions in the phosphors sample are uniformly distributed in the matrix.

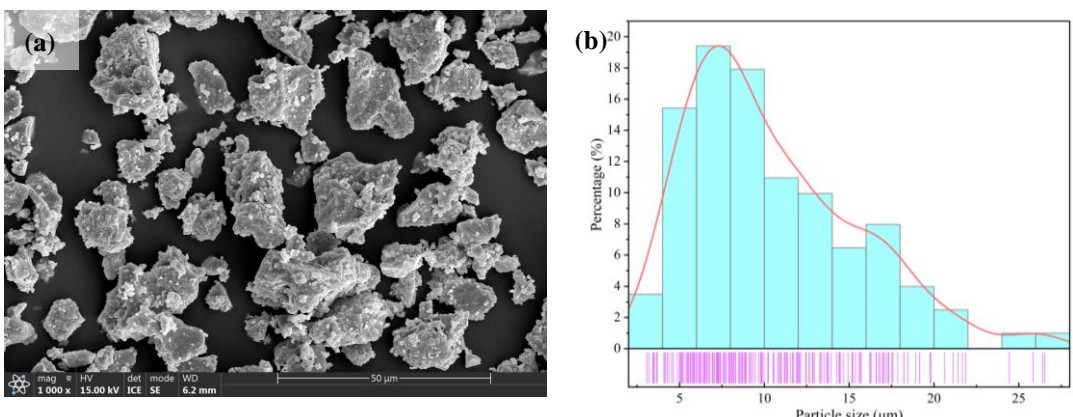

**Figure 2.** The SEM microtopography (**a**) and particle size distribution (**b**) of $K_5La_{0.93}(MoO_4)_4$: $0.07Sm^{3+}$ phosphors.

### 3.2. Fluorescence Property

Figure 3a displays the excitation and emission spectra of $K_5La_{0.93}(MoO_4)_4$: $0.07Sm^{3+}$ phosphors. Under the monitoring of 600 nm emission, the sample exhibits multiple excita-

tion peaks of 343 nm, 359 nm, 373 nm, 401 nm, 436 nm and 472 nm, which correspond to the $^6H_{2/5}{\rightarrow}^4K_{17/2}$, $^4H_{7/2}$, $^6P_{7/2}$, $^4F_{7/2}$, $^4G_{9/2}$, and $^4I_{11/2}$ transitions of $Sm^{3+}$ ion respectively. Under 401 nm excitation, the phosphors exhibit obvious fluorescence emission peaking at 561 nm, 600 nm and 646 nm, attributed to the $^4G_{5/2}{\rightarrow}^6H_{5/2}$, $^6H_{7/2}$ and $^6H_{9/2}$ transitions of $Sm^{3+}$ ion severally. In addition, a weaker emission peak at 705 nm is observed, ascribed to the $^6G_{2/5}{\rightarrow}^6H_{11/2}$ transition of the $Sm^{3+}$ ions [26,27].

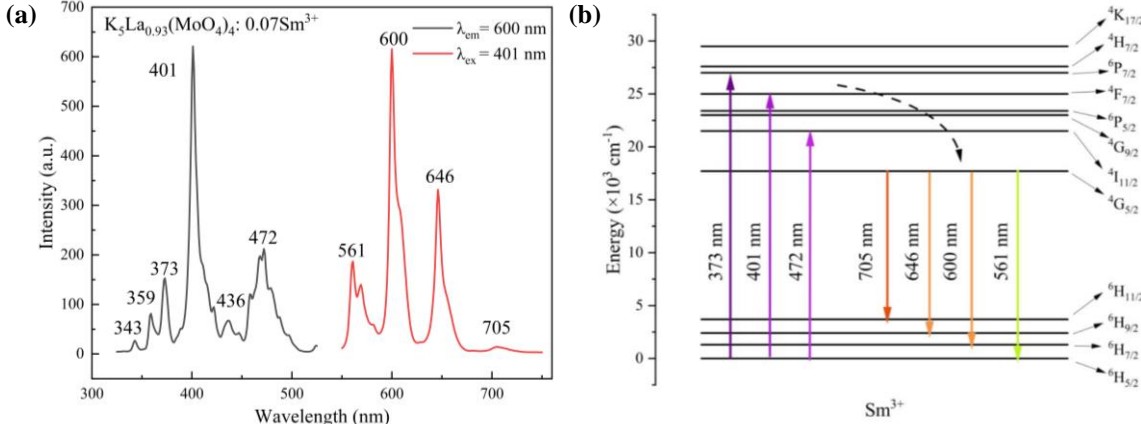

**Figure 3.** Excitation (black) and emission (red) spectra of $K_5La_{0.93}(MoO_4)_4$: $0.07Sm^{3+}$ phosphors (**a**) and energy levels diagram of $Sm^{3+}$ ion (**b**).

Figure 3b shows the schematic diagram of the energy levels of $Sm^{3+}$ ions. After excited, the electrons in the ground state absorb energy and transition to the excited state, and then relax to $^4G_{5/2}$ through non-radiation (NR). Finally, these electrons transition back to the ground state, emitting the corresponding wavelength of visible light.

Figure 4 exhibits the emission spectra of $K_5La_{1-x}(MoO_4)_4$: $xSm^{3+}$ phosphors, and the inset presents the variation trend of the emission intensity at 600 nm ($^4G_{5/2}{\rightarrow}^6H_{7/2}$) with doping concentrations (x). Obviously, when the doping concentration is relatively small, the emission intensity of the phosphors is enhanced due to the increase of the luminous centers, reaching a maximum value at x = 0.07, and then concentration quenching occurs.

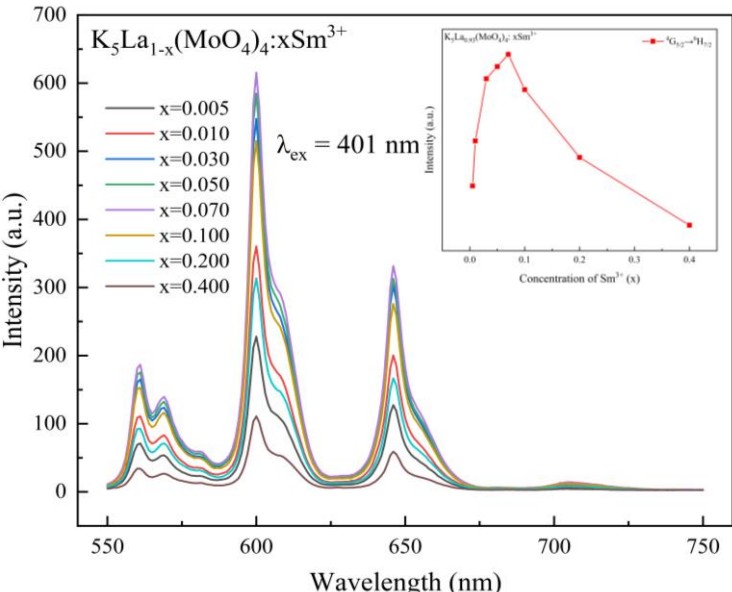

**Figure 4.** Emission spectra of $K_5La_{1-x}(MoO_4)_4$: $xSm^{3+}$ phosphors and the emission intensity at 600 nm change with $Sm^{3+}$ doping concentrations (x) (inset).

In general, the concentration quenching mainly takes place in three ways: radiation reabsorption, exchange and multi-dipole interaction [28]. Radiation reabsorption requires obvious overlap between the excitation spectra and emission spectra of phosphors [29], which is not observed in Figure 4, thus radiation reabsorption is not considered. The exchange is a short-range interaction, which plays a role when the critical distance ($R_c$) of energy transfer between the two neighboring luminous center ions is shorter than 5 Å [30]. The smaller the critical distance, the less probability of concentration quenching. According to Blass's studies [31,32], $R_c$ can be approximately equal to twice the radius of the sphere in the acting volume:

$$R_c \approx 2(3V/4\pi N\mathrm{x})^{1/3} \tag{1}$$

where $V$ is the volume of unit cell, $N$ is the number of replaceable atoms in the unit cell, and x refers to the critical quenching concentration. According to Formula (1), the $R_c$ of $K_5La(MoO_4)_4$ matrix is calculated to be 14.30 Å, much longer than 5 Å. In other words, exchange does not contribute to the concentration quenching. Therefore, multi-dipole interaction can be deduced to be responsible for the concentration quenching in $K_5La(MoO_4)_4:Sm^{3+}$ phosphors.

Dexter's research [33,34] indicates that the multi-dipole interaction between the luminescent centers in phosphors could be described by the following formula:

$$\log (I/\mathrm{x}) = A - \theta/3\log\mathrm{x} \tag{2}$$

where I stands for the emission intensity, x is the concentration of the luminescent centers, and A is a constant. When $\theta$ is equal to 6, 8, 10, it corresponds to dipole-dipole, dipole-quadrupole and quadrupole-quadrupole interaction separately.

Figure 5 exhibits the relationship between lg(I/x) and lg(x) in the $K_5La_{1-x}(MoO_4)_4$: $xSm^{3+}$ phosphors according to the Formula (2), which is linearly fitted with the slope ($-\theta/3$) calculated as $-1.97$, i.e., the $\theta$ is equal to 6, indicating the dipole-dipole interaction plays a role.

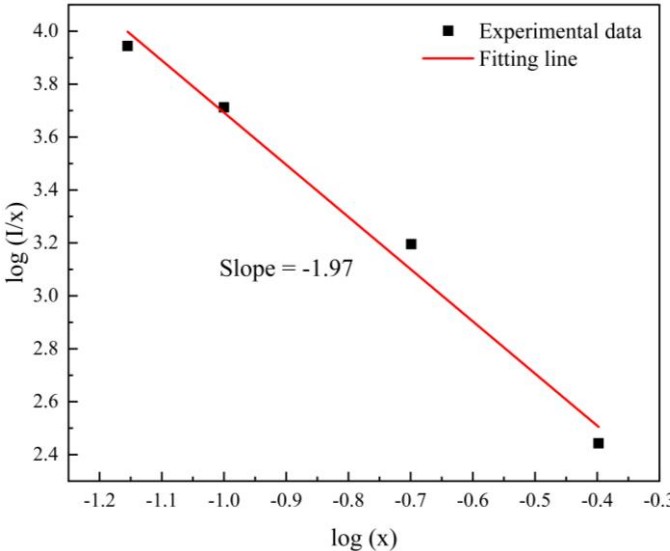

**Figure 5.** The relationship between log(I/x) and log(x) in $K_5La_{1-x}(MoO_4)_4$: $xSm^{3+}$ phosphors.

Figure 6 exhibits the curves of lifetime decay for $K_5La_{1-x}(MoO_4)_4:xSm^{3+}$ phosphors. Using quadratic exponents to fit the fluorescence lifetime decay curves is a common analytical method [35,36]:

$$I(t) = I_0 + A_1 \exp(-t/t_1) + A_2 \exp(-t/t_2) \tag{3}$$

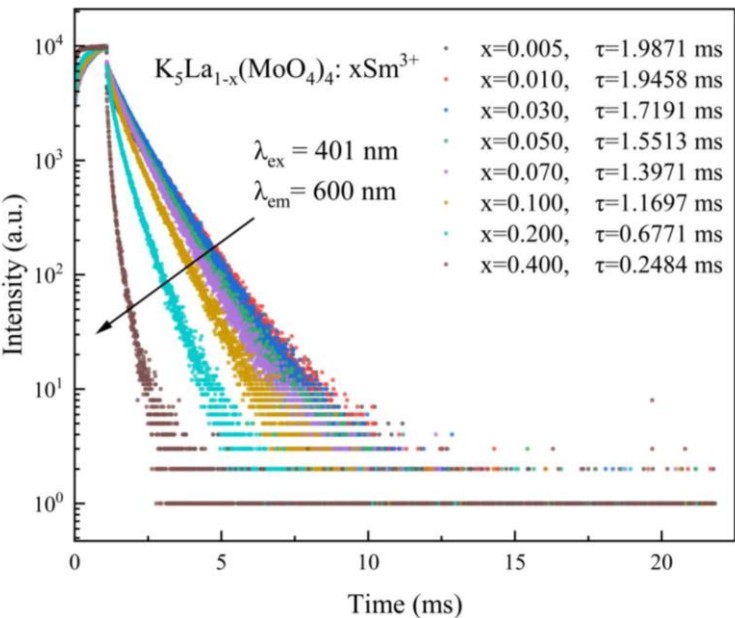

**Figure 6.** Fluorescence lifetime test results of $K_5La_{1-x}(MoO_4)_4$: $xSm^{3+}$ phosphors.

In which $I_0$ and $I(t)$ are respectively the initial emission intensity and the emission intensity after the decay time t, $A_1$ and $A_2$ are constants, and the $t_1$ and $t_2$ are two attenuation life factors. Based on this, the average decay life ($\tau$) of the phosphors can be calculated by the following formula [37] and listed in Figure 6:

$$\tau = (A_1t_1^2 + A_2t_2^2)/(A_1t_1 + A_2t_2) \tag{4}$$

On the whole, the fluorescence lifetime of phosphors decreases with the increase of doping concentration, which might be caused by the interaction or energy transfer between $Sm^{3+}$ ions [38,39].

As well known, the luminescence properties of rare earth ions are affected by surrounding crystal field environment. In order to further improve the luminescence properties of $K_5La(MoO_4)_4$: $Sm^{3+}$ phosphors, $Ln^{3+}$ (Ln = Sc, Y or Gd) ions are co-doped in $K_5La_{0.93}(MoO_4)_4$: $0.07Sm^{3+}$ phosphors although they do not emit visible light.

As can be seen from Figure S3, the shape and peaking position of XRD patterns do not change significantly after co-doping with $Ln^{3+}$ ions although deviation of peaks is observed to some different degree, which could be due to the radii difference between $Ln^{3+}$ ions and $La^{3+}$ ions, as shown in Table 2, where $D_r$ represents the difference between ions and is calculated by the following formula:

$$D_r = |(R_m - R_d)/R_m| \times 100\% \tag{5}$$

**Table 2.** Comparison of radii between $Ln^{3+}$ ions and $La^{3+}$ ion.

| Cation | Radius (Å) (CN = 6) | $D_r$ (%) |
|:------:|:------:|:------:|
| $La^{3+}$ | 1.032 | \ |
| $Sm^{3+}$ | 0.958 | 7.17 |
| $Sc^{3+}$ | 0.745 | 27.8 |
| $Y^{3+}$ | 0.900 | 12.8 |
| $Gd^{3+}$ | 0.938 | 9.11 |

In Formula (5), $R_m$ is the radius of $La^{3+}$ ions and $R_d$ is the radius of different $Ln^{3+}$ ions. Studies have demonstrated that when $D_r$ is less than 30%, the dopant ions are more inclined to enter the matrix site in the way of replacement solution [40].

In terms of microtopography, as shown in Figure S4, the co-doping of $Ln^{3+}$ ions does not significantly change the morphology of phosphors. The Mapping results in Figures S5–S7 further indicate that $Ln^{3+}$ ions, similar to $Sm^{3+}$ ions, are uniformly distributed in the phosphors as dopant.

Figure 7 provides the excitation and emission spectra of $Ln^{3+}$ co-doping phosphors. It can be found that the shape and peaking position of excitation and emission spectra have not changed significantly after co-doping of different $Ln^{3+}$ ions. However, the emission intensity of phosphors is improved to a certain extent.

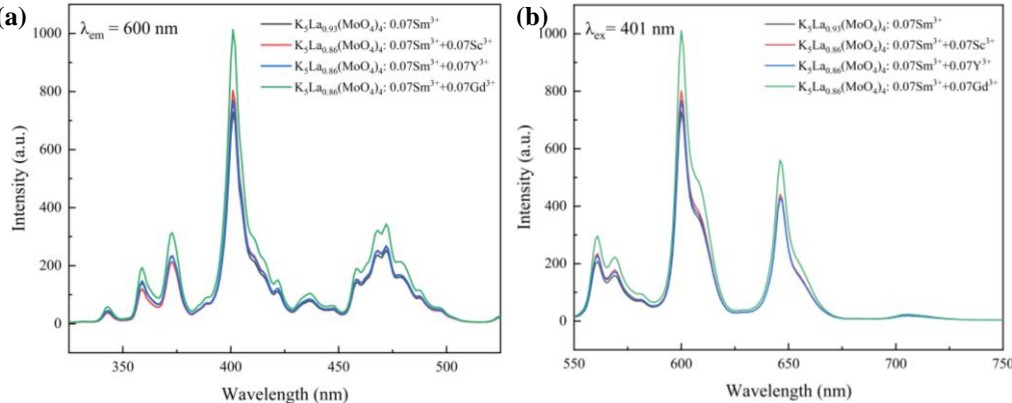

**Figure 7.** The excitation (**a**) and emission (**b**) spectra of $K_5La_{0.86}(MoO_4)_4$: $0.07Sm^{3+}$, $0.07Ln^{3+}$ (Ln = Sc, Y and Gd) phosphors.

Figure 8 shows the change of main emission peaks intensity after $Ln^{3+}$ ions are co-doped and exhibits its normalization analysis. It can be seen more directly that all the three kinds of $Ln^{3+}$ ions have either large or small enhancement effects on the three emission peaks intensity. Taking the strongest emission peak of 600 nm as an example, after $Sc^{3+}$, $Y^{3+}$ or $Gd^{3+}$ ions are co-doped, the emission intensity increases to 1.102, 1.055 and 1.387 times of the original value of $K_5La_{0.93}(MoO_4)_4$: $0.07Sm^{3+}$ respectively. Due to the certain ionic radius difference between $Ln^{3+}$ ions and $La^{3+}$ ions, the cell would produce a certain degree of contraction, which would enhance the crystal field strength surrounding $Sm^{3+}$ ions [41]. According to the crystal field theory, the luminescence intensity of rare earth ions would increase with the increase of the crystal field intensity [25,42,43].

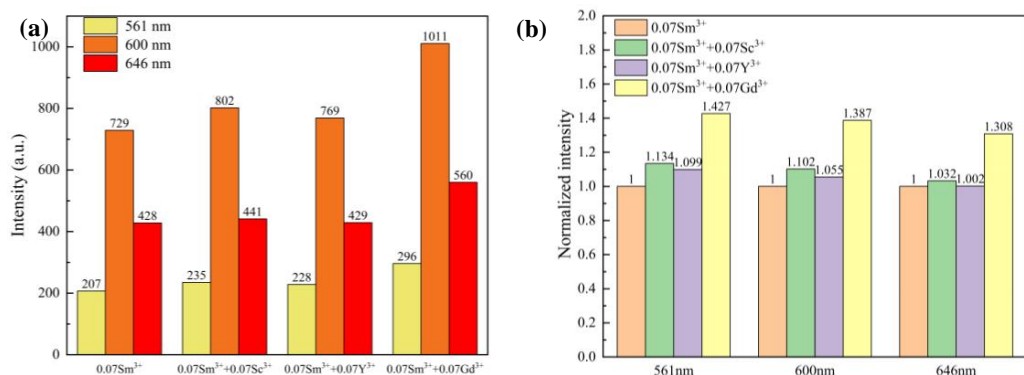

**Figure 8.** The effect of $Ln^{3+}$ ions co-doping on the intensity of emission peaks (**a**) and the normalized intensity of main emission peaks (**b**) after $Ln^{3+}$ ions co-doping.

According to Table 2, the order of radius difference between $Ln^{3+}$ ions and $La^{3+}$ ion is $D_r(Sc^{3+}) > D_r(Y^{3+}) > D_r(Gd^{3+})$, so the $Sc^{3+}$ ions co-doped phosphors should have the strongest gain effect. However, the results indicate that the gain effect of $Ln^{3+}$ ions on phosphors is $Gd^{3+} > Sc^{3+} > Y^{3+}$, which could be explained by the energy transfer between $Gd^{3+}$ and $Sm^{3+}$ ions [44,45].

Figure 9 is the schematic diagram of the energy transfer between $Gd^{3+}$ and $Sm^{3+}$ ions. Although $Sc^{3+}$, $Y^{3+}$ and $Gd^{3+}$ are all categorized as inert rare earth ions that do not emit light in the visible region [46–48], unlike empty filled $4f^0$ for $Sc^{3+}$ and $Y^{3+}$ ions, the 4f electron orbitals of $Gd^{3+}$($4f^7$) are half filled, which allows energy transfer through f-f leaps from $Gd^{3+}$ ions to luminescent center ions [49,50]. In the case of $Gd^{3+}$ ions being excited, electrons transition from the ground state ($^8S_{7/2}$) to the excited state ($^6D_j$), then relax to the lower energy levels ($^6I_J$, $^6P_j$) and transfer to the excited states of $Sm^{3+}$. After further relax to the $^4G_{5/2}$ level of $Sm^{3+}$ ions, the electrons eventually transition back to the ground state and emit light. In this way, co-doping of $Gd^{3+}$ ions increases the number of electrons transitioned from the excited state to the ground state through this energy transfer (ET) process and enhances the luminescence intensity. In order to describe this energy transfer in more detail, we characterized the fluorescence lifetime of $K_5La_{0.86}(MoO_4)_4$: $0.07(Sm^{3+}, Ln^{3+})$ phosphors.

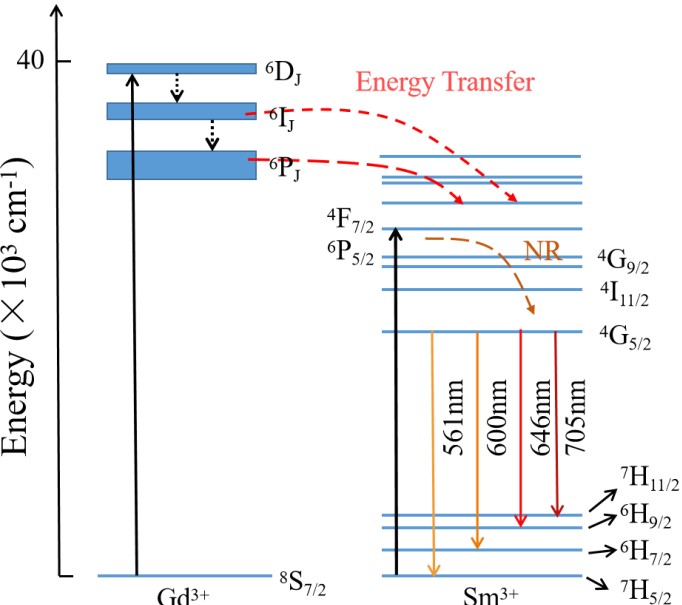

**Figure 9.** Energy transfer diagram between $Gd^{3+}$ and $Sm^{3+}$ ions.

Figure 10 displays the curves of fluorescence lifetime of $K_5La_{0.86}(MoO_4)_4$: $0.07Sm^{3+}$, $0.07Ln^{3+}$ (Ln = Sc, Y and Gd) phosphors. It can be seen that the fluorescence lifetime does not change significantly after $Sc^{3+}$ or $Y^{3+}$ ions are co-doped, while the fluorescence lifetime of phosphors increases from 1.3971 ms to 1.7027 ms after co-doping of $Gd^{3+}$ ions. The increase of fluorescence lifetime confirms the energy transfer of $Gd^{3+}$ ions to $Sm^{3+}$ ions to some extent [51–53].

Figure 11 expresses the emission spectra of $K_5La_{0.93}(MoO_4)_4$: $0.07Sm^{3+}$ (a) and $K_5La_{0.86}(MoO_4)_4$: $0.07Sm^{3+}$, $0.07Gd^{3+}$ (b) phosphors, as well as the changes of emission intensity at different temperature (inset). With the increase of temperature, the emission intensity of phosphors first shows a certain enhancement and reaches the highest value at 348 K, and then with the further increase of temperature, although the emission intensity of phosphor gradually decreases, it still keeps at 1.3–1.9 times of the initial emission intensity at 473 K.

In theory, with the increase of temperature, thermal motion of the molecules becomes more active, the phonon action is enhanced, and the probability of the excited electrons returning to the ground state through the form of non-radiative transition increases, eventually the thermal quenching takes place [54]. However, for $K_5La_{0.93}(MoO_4)_4$: $0.07Sm^{3+}$ and $K_5La_{0.86}(MoO_4)_4$: $0.07Sm^{3+}$, $0.07Gd^{3+}$ phosphors, the emission intensity increases instead of decreasing, showing an abnormal thermal quenching phenomenon, i.e., zero thermal quenching is found. As a matter of the fact, the thermal stability of a phosphors can be

expressed by the thermal activation energy $E_a$, which can be calculated by the Arrhenius formula below [55]:

$$\ln(I_0/I - 1) = \ln A - E_a/kT \tag{6}$$

where $I_0$ and $I$ are the emission intensity at room temperature and T respectively, and k is the Boltzmann constant.

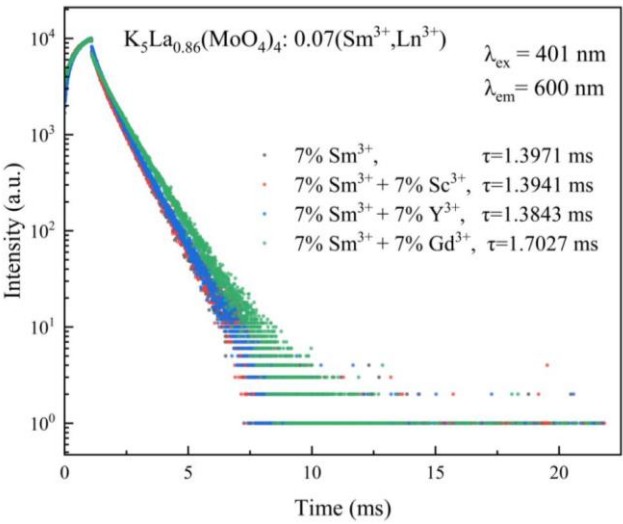

**Figure 10.** Fluorescence lifetime test results of $K_5La_{0.86}(MoO_4)_4$: $0.07Sm^{3+}$, $0.07Ln^{3+}$ (Ln = Sc, Y and Gd) phosphors.

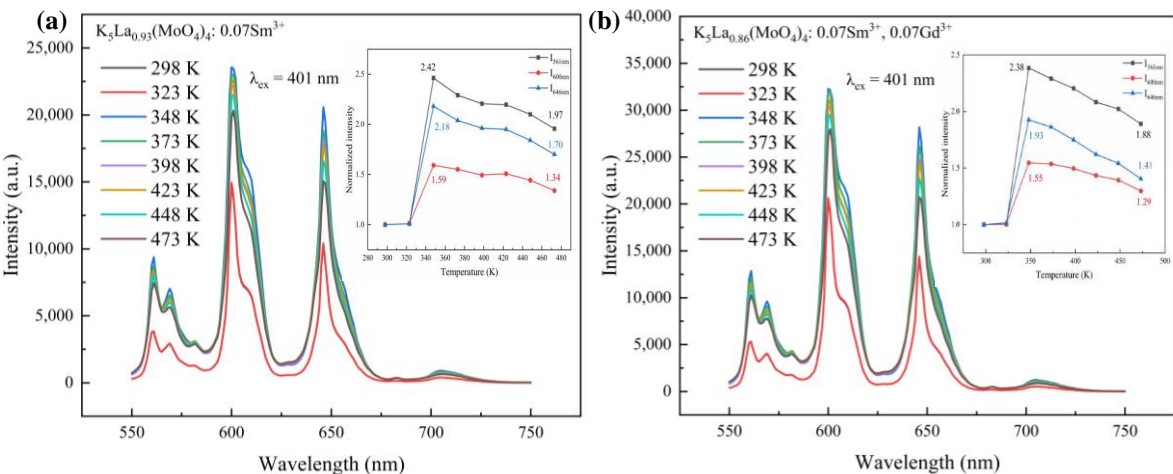

**Figure 11.** Emission spectra at different temperatures of $K_5La_{0.93}(MoO_4)_4$: $0.07Sm^{3+}$ (**a**) and $K_5La_{0.86}(MoO_4)_4$: $0.07Sm^{3+}$, $0.07Gd^{3+}$ (**b**) phosphors, and the relationship between emission intensity and temperature (insets).

According to the fitting results in Figure 12, the thermal activation energy $E_a$ of $K_5La_{0.93}(MoO_4)_4$: $0.07Sm^{3+}$ and $K_5La_{0.86}(MoO_4)_4$: $0.07Sm^{3+}$, $0.07Gd^{3+}$ phosphors is calculated as 0.28 eV and 0.27 eV respectively.

Taken into account the temperature range (298–473 K) of characterization, the possibility of phase transition is little, and far fewer studies reported that $Sm^{3+}$ ions have thermal coupling energy levels (TCLs) like $Dy^{3+}$ or $Er^{3+}$ ions [56,57]. This anomalous thermal quenching phenomenon of the target phosphors might be explained by the electron traps [58–61].

Figure 13 is a schematic diagram of electron trapping. In general, after excitation, the electrons transition from the ground state to the excited state through the process of ①, then relax to the excited state level with lower energy through ②, and finally return to the

ground state via ③ and emit visible light. However, once the electron traps exist in the phosphors, a part of the excited electrons would be captured by them (④) and cannot get enough energy to escape. When the temperature is high enough, these electrons would absorb heat energy and (⑤) escape from the traps to reach the excited state, and finally transition back to the ground state via radiation (⑥). In order to confirm the existence of electron traps in phosphors, the TL spectrum of phosphors are characterized.

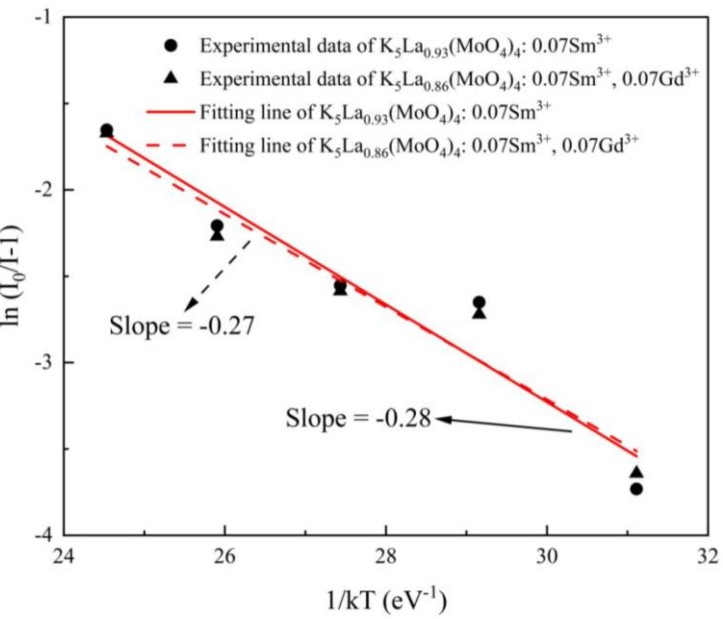

**Figure 12.** The relationship between $\ln(I_0/I-1)$ and $1/kT$ in $K_5La_{0.93}(MoO_4)_4$: $0.07Sm^{3+}$ and $K_5La_{0.86}(MoO_4)_4$: $0.07Sm^{3+}$, $0.07Gd^{3+}$ phosphors.

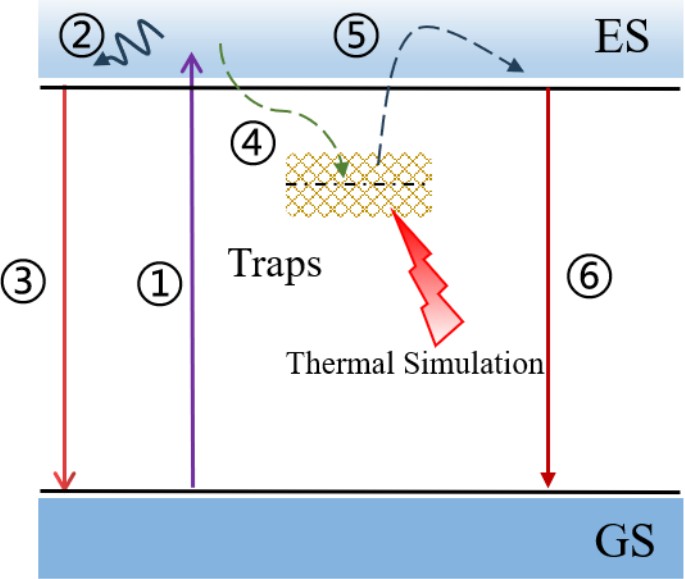

**Figure 13.** Schematic diagram of the electronic traps.

Figure 14 is the TL spectrum of the $K_5La_{0.93}(MoO_4)_4$: $0.07Sm^{3+}$ and $K_5La_{0.86}(MoO_4)_4$: $0.07Sm^{3+}$, $0.07Gd^{3+}$ phosphors, from which it can be told that the TL spectra have peaks centered at 405 K and 406 K in the range of 300–500 K, at a heating rate of 1 K/s. The appearance of TL peak proves the existence of electron traps to some extent. To better describe the relationship between the electron traps and the TL spectrum, the following

two formulas could be used to calculate the depth ($E_{trap}$) and density ($N_0$) of the electron traps in the phosphors [35,62]:

$$E_{trap} = T_m/500 \tag{7}$$

$$N_0 = \omega \times I_m/\{\beta \times [2.52 + 10.2 \times (\mu_g - 0.42)]\} \tag{8}$$

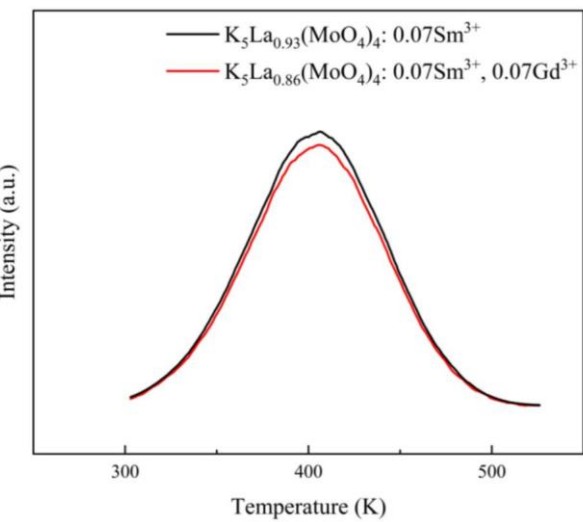

**Figure 14.** TL spectrum of the $K_5La_{0.93}(MoO_4)_4$: $0.07Sm^{3+}$ and $K_5La_{0.86}(MoO_4)_4$: $0.07Sm^{3+}$, $0.07Gd^{3+}$ phosphors.

In the above formulas, $T_m$ is the temperature corresponding to the highest point of the TL peak, and $\omega$ is defined as the shape parameter, $\omega = \tau + \delta$, in which $\tau$ is the low-temperature half-width, and $\delta$ is the high temperature half-width. The symmetry parameter $\mu_g = \delta/(\tau + \delta)$, $\beta$ is the heating rate, and $I_m$ is the intensity of the TL peak. According to these formulas, the depth and density of the electron trap in the phosphors are 0.81 eV, 0.812 eV and $2.15 \times 10^6$, $2.09 \times 10^6$ respectively. The $E_{trap}$ and $N_0$ of $K_5La_{0.93}(MoO_4)_4$: $0.07Sm^{3+}$ phosphors are slightly larger than that of $Gd^{3+}$ co-doped phosphors, which is consistent with the thermal stability they exhibit in the illustrations in Figure 11.

Quantum efficiency is another important index to evaluate the performance of phosphors. Figure 15 presents the tested diagram of quantum efficiency of $K_5La_{0.86}(MoO_4)_4$: $0.07Sm^{3+}$, $0.07Gd^{3+}$ phosphors. The internal quantum efficiency of phosphors could be calculated from the ratio of the emission to the absorption peak area of the sample [63,64]:

$$\eta_q = \left(\int_{LS} - \int_{LB}\right)/\left(\int_{EB} - \int_{ES}\right) \tag{9}$$

In Formula (9), $\int_{LS}$ and $\int_{LB}$ represent the emission spectral area of phosphors and blank sample respectively, while the $\int_{EB}$ and $\int_{ES}$ represent the absorption spectral area of phosphors and blank sample. According to Formula (9), the quantum efficiency of $K_5La_{0.86}(MoO_4)_4$: $0.07Sm^{3+}$, $0.07Gd^{3+}$ phosphors is calculated to be 74.63%.

Color coordinates are an intuitive representation of the luminous color of the phosphors. Figure 16 displays the color coordinates of the $K_5La_{1-x}(MoO_4)_4$: $xSm^{3+}$ and $K_5La_{0.86}(MoO_4)_4$: $0.07Sm^{3+}$, $0.07Ln^{3+}$ (Ln = Sc, Y or Gd) (b) phosphors, while Figures S8 and S9 indicate the actual fluorescence photos of these two series of phosphors under ultraviolet irradiation, which suggest the color coordinates of the samples are all located in the orange-red region. Meanwhile, Tables 3 and 4 list the color coordinates (CIE), color temperature (CCT) and color purity of phosphors samples with different doping concentration, revealing the color temperature of all the phosphors is about 1750–1760 K, and it's worth noting that the color purity is as high as 99.97%.

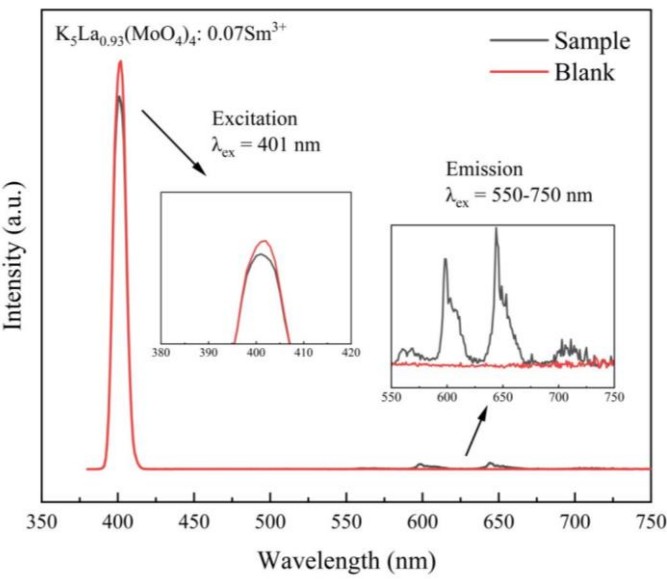

**Figure 15.** Quantum efficiency test diagram of $K_5La_{0.86}(MoO_4)_4$: $0.07Sm^{3+}$, $0.07Gd^{3+}$ phosphors.

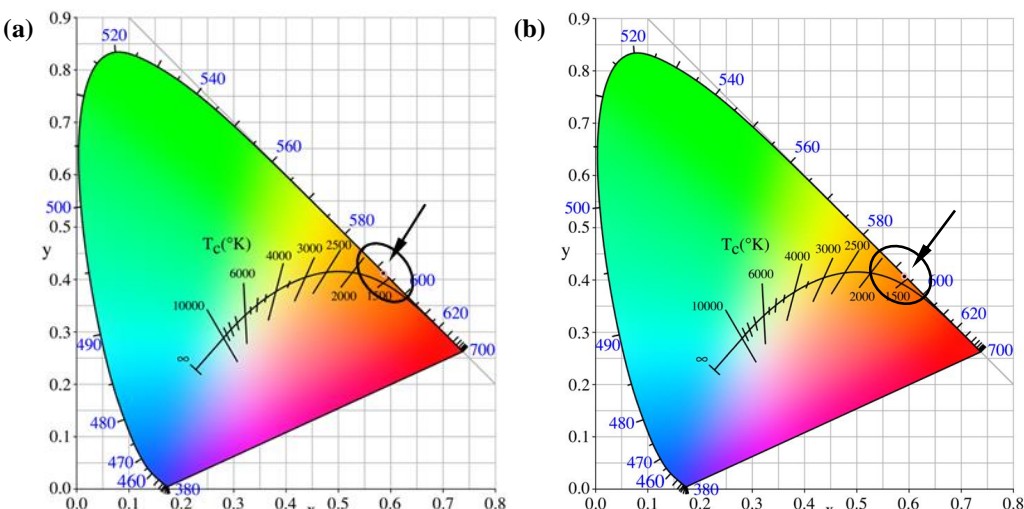

**Figure 16.** Color coordinates (CIE) of $K_5La_{1-x}(MoO_4)_4$: $xSm^{3+}$ (**a**) and $K_5La_{0.86}(MoO_4)_4$: $0.07Sm^{3+}$, $0.07Ln^{3+}$ (Ln = Sc, Y and Gd) (**b**) phosphors.

**Table 3.** The CIE, CCT and color purity of $K_5La_{1-x}(MoO_4)_4$: $xSm^{3+}$ phosphors.

| Number | x | CIE ($x_c$, $y_c$) | CCT (K) | Color Purity (%) |
|---|---|---|---|---|
| 1 | 0.005 | (0.5880, 0.4113) | 1764 | 99.97 |
| 2 | 0.01 | (0.5888, 0.4105) | 1761 | 99.97 |
| 3 | 0.03 | (0.5895, 0.4098) | 1759 | 99.97 |
| 4 | 0.05 | (0.5896, 0.4097) | 1759 | 99.97 |
| 5 | 0.07 | (0.5898, 0.4095) | 1758 | 99.97 |
| 6 | 0.10 | (0.5899, 0.4094) | 1758 | 99.97 |
| 7 | 0.20 | (0.5894, 0.4099) | 1759 | 99.97 |
| 8 | 0.40 | (0.5870, 0.4123) | 1767 | 99.97 |

**Table 4.** The CIE, CCT and color purity of $K_5La_{0.86}(MoO_4)_4$: $0.07Sm^{3+}$, $0.07Ln^{3+}$ (Ln = Sc, Y or Gd) phosphors.

| Number | Dopant | CIE ($x_c$, $y_c$) | CCT (K) | Color Purity (%) |
|--------|--------|--------------------|---------|------------------|
| 1 | $0.07Sm^{3+}$ | (0.5898, 0.4095) | 1758 | 99.97 |
| 2 | $0.07Sm^{3+} + 0.07Sc^{3+}$ | (0.5922, 0.4071) | 1753 | 99.97 |
| 3 | $0.07Sm^{3+} + 0.07Y^{3+}$ | (0.5919, 0.4074) | 1754 | 99.97 |
| 4 | $0.07Sm^{3+} + 0.07Gd^{3+}$ | (0.5924, 0.4069) | 1753 | 99.97 |

## 4. Conclusions

In this work, a series of $K_5La_{1-x}(MoO_4)_4$: $xSm^{3+}$ and $K_5La_{0.86}(MoO_4)_4$: $0.07Sm^{3+}$, $0.07Ln^{3+}$ (Ln = Sc, Y or Gd) red phosphors were synthesized via solid state reaction at 600 °C. The excitation and emission spectra of the samples suggest that the phosphors can be efficiently excited at 401 nm and emit visible light at 561 nm, 600 nm and 646 nm, which are corresponding to the $^4G_{5/2} \rightarrow {}^6H_{j/2}$ (j = 5, 7 and 9) energy transitions of $Sm^{3+}$ ion with the highest intensity at x = 0.07, and the concentration quenching is attributed to the dipole-dipole interaction. The emission intensity of the phosphors is further improved by co-doping of $Ln^{3+}$ (Ln = Sc, Y or Gd) ions in the same amount via the enhanced the crystal field strength surrounding $Sm^{3+}$ ions, among which $Gd^{3+}$ ions exhibit the best gain effect due to their energy transfer to $Sm^{3+}$ ions, and the emission intensity is increased by 1.3–1.4 times. It is interesting that $K_5La_{1-x}(MoO_4)_4$: $xSm^{3+}$ and $K_5La_{0.86}(MoO_4)_4$: $0.07Sm^{3+}$, $0.07Ln^{3+}$ phosphors have excellent thermal stability, even an abnormal thermal quenching phenomenon, i.e., a zero thermal quenching appears. At 473 K, the emission intensity of $K_5La_{0.86}(MoO_4)_4$:$0.07Sm^{3+}$, $0.07Gd^{3+}$ phosphor is still 1.3–1.8 times that at room temperature, which can be attributed to the electron traps in the phosphors. It is worth mentioning that under 401 nm excitation, the quantum efficiency of the sample is 74.63%. At the same time, the color temperature of the phosphors is 1500–1600 K; and the color purity is as high as 99.97%. These results imply that this red phosphor has potential application in the LED field.

**Supplementary Materials:** The following supporting information can be downloaded at: https://www.mdpi.com/article/10.3390/inorganics11100406/s1, Figure S1: Schematic diagram of $K_5La(MoO_4)_4$ crystal structure from different views (a) normal, (b) a axis, (c) b axis, (a) c axis; Figure S2: The Mapping patterns of $K_5La_{0.93}(MoO_4)_4$: $0.07Sm^{3+}$ sample: entirety (a), Mo (b), K (c), O (d), La (e), Sm (f); Figure S3: The XRD pattern of $K_5La_{0.86}(MoO_4)_4$: $0.07(Sm^{3+}, Ln^{3+})$ phosphors; Figure S4: The SEM photos of $K_5La_{0.86}(MoO_4)_4$: $0.07Sm^{3+}$, $0.07Ln^{3+}$ (Ln = Sc (a), Y (b) and Gd (c)) phosphors; Figure S5: The EDS and Mapping patterns of $K_5La_{0.86}(MoO_4)_4$: $0.07(Sm^{3+}, Sc^{3+})$ phosphors (inset a-f): Sm (a), Mo (b), K (c), O (d), La (e), Sc (e); Figure S6: The EDS and Mapping patterns of $K_5La_{0.86}(MoO_4)_4$: $0.07(Sm^{3+}, Y^{3+})$ phosphors (insets a-f): Sm (a), Mo (b), K (c), O (d), La (e), Y (e); Figure S7: The EDS and Mapping patterns of $K_5La_{0.86}(MoO_4)_4$: $0.07(Sm^{3+}, Gd^{3+})$ phosphors (insets a-f): Sm (a), Mo (b), K (c), O (d), La (e), Gd (e); Figure S8: Actual fluorescence photos of $K_5La_{1-x}(MoO_4)_4$: $xSm^{3+}$, x= 0.005 (a), 0.01 (b), 0.03 (c), 0.05 (d), 0.07 (e), 0.10 (f), 0.20 (g) and 0.40 (h); Figure S9: Actual fluorescence photos of $K_5La_{0.86}(MoO_4)_4$: $0.07Sm^{3+}$, $0.07Ln^{3+}$ (Ln = Sc, Y and Gd) phosphors: Sm (a), Sm+Sc (b), Sm+Y (c) and Sm+Gd (d).

**Author Contributions:** Conceptualization, methodology, T.Z.; validation, T.Z., S.Z. and D.Z.; investigation, S.Z.; resources, D.Z.; data curation, T.Z.; writing—original draft preparation, T.Z.; writing—review and editing, T.Z.; visualization, T.Z.; supervision, D.Z.; project administration, D.Z.; All authors have read and agreed to the published version of the manuscript.

**Funding:** This research received no external funding.

**Data Availability Statement:** The data presented in this study are available on request from the corresponding author.

**Conflicts of Interest:** The authors declare no conflict of interest.

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
