# Peer review of "A Novel Zero-Thermal-Quenching Red Phosphor with High Quantum Efficiency and Color Purity"

_inorganics, doi:10.3390/inorganics11100406_

Round 1
Reviewer 1 Report
Review Report:
In their manuscript Zhao et al. demonstrated the synthesis, characterization and the detailed emission properties of a series of red phosphors. The concentration-dependent quenching mechanism involving Sm3+ is ascribed to dipole-dipole interactions. Their results further show that Ln3+ co-doping increases the emission intensity.
This work is fundamentally important for the next generation LEDs and fits with the journal scope. However, there are some modifications required. My specific comments are copied below. I use the following abbreviations, P-page number and L-line number.
Specific Comments:
1. P4-L108: The line color for the excitation and emission spectra is opposite with respect to the figure legends.
2. P4-Figure 3a: I encourage the authors to put y-axis values (labels) for better readability.
3. P4-Figure 4: Y-axis labels would provide a good understanding of the quenching.
4. P4-Figure 4: Provide an integrated absolute difference spectra around 600 nm as a function of doping concentration instead of peak intensities for better understanding of the quenching effect.
5. P5-L154: It should be log not lg both in the text and in figure 5 axes titles.
6. P7-Figure 7: Provide y-axis values (labels) for the comparison.
Author Response
Thank you for your comments, all the point-to-point responses have been attached to the word-file below.

Reviewer 2 Report
The following issues must be addressed:
1. Introduction part must be improved and must underline what is new and innovative in this work compared with other papers.
2. Provide the diffraction planes.
3. Discuss if there are any preferential growth.
4. The particle size is not between 10-20 um. Please provide higher resolution images.
5. EDS is not relevant if it is made on one particle; additionally, is not clear how authors decide that there is no segregation.
6. There must be more correlations between analysis.
Author Response

(The authors gave the same response as above.)

Reviewer 3 Report
The authors describe in this paper a series of different red phosphors and an exhaustive characterization of them. The work is well written, and the conclusions are consistent with the experimental results that the authors show.
The article is in the scope of the journal. I would like to point out the high number or references that support the different statements in the work and the high quality figures 9, 13 and 16 to make clearer the lecture for the reader. For all these things and for the consistent experimental results I consider the article suitable for publication. Only minor revisions are needed:
1. I wonder if it is possible another different procedure for the synthesis than 600ºC during 8 hours. It is a very high temperature for a long time. Have you tested other different methodologies?
2. In the work, the authors compare for example in Figure 1b, experimental results with theoretical ones. However, I miss a deeper explanation about the theoretical calculations that the authors have carried out.
3. Please, in the text when the authors describe any figure, delete the point after the figure. Write Figure 3a, not Figure.3a.
After these minor suggestions, I consider that the article can be published in Inorganics.
Author Response

(The authors gave the same response as above.)

Round 2
Reviewer 2 Report
The manuscript can be published in present form.